# Maturation of the Goat Rumen Microbiota Involves Three Stages of Microbial Colonization

**DOI:** 10.3390/ani9121028

**Published:** 2019-11-25

**Authors:** Ke Zhang, Bibo Li, Mengmeng Guo, Gongwei Liu, Yuxin Yang, Xiaolong Wang, Yulin Chen, Enping Zhang

**Affiliations:** 1College of Animal Science and Technology, Northwest Agriculture and Forestry University, Yangling District, Xianyang 712100, China; kezhang@nwafu.edu.cn (K.Z.); libibo1988@126.com (B.L.); liugongwei2010@163.com (G.L.); yangyuxin2002@126.com (Y.Y.); xiaolongwang@nwafu.edu.cn (X.W.); 2College of Veterinary Medicine, Northwest Agriculture and Forestry University, Yangling District, Xianyang 712100, China; mmengguo@163.com

**Keywords:** microbiome, ruminant, rumen, fungi, pre-weaning model, 16S ribosomal DNA

## Abstract

**Simple Summary:**

Considerable attention has recently been focused on the rumen microbiome, which has been implicated in regulating a ruminant’s nutrient metabolism. From birth onwards, the colonization of the rumen microbial community is thus of crucial importance for growth and fiber digestion of goats. In this study, we have provided details of the progression of changes and colonization of ruminal bacteria and fungi before weaning. We have also predicted the molecular functions of the bacterial microbiota using CowPi. Our finding confirmed that maturation of the goat rumen microbiota involves three stages of core microbial colonization. The study of rumen microbial of young ruminants will benefit the optimization of feeding strategies to promote the development and digestion of a healthy rumen microbiota in later life.

**Abstract:**

With increasing age, the rumen microbiota of new-born ruminants become central in the translation of fibrous feed substances into essential nutrients. However, the colonization process of the microbial community (especially fungal community) remains poorly understood in ruminants at pre-weaning stages. In this study, the rumen bacterial and fungal colonization processes were investigated in goats at eight stages using amplicon sequencing. For bacteria, we found 36 common core genera at D0, D3, D14, D28, and D56, including mainly *Bacillus*, *Alloprevotella*, *Bacteroides*, *Prevotella_1*, *Lactococcus,* and *Ruminococcaceae_NK4A214*. Firmicutes was the dominant phylum among the total microbiota in newborn goat kids (prior to nursing), while *Bacillus*, *Lactococcus*, and *Pseudomonas* were predominant genera. Interestingly, the proportion of *Bacillus* was as high as 55% in newborn animals. After milk nursing, the predominant phylum changed to Bacteroidetes, while the proportion of *Bacillus* and *Lactobacillus* was very low. CowPi was used to predict the functional gene pathways and we found increases in the abundance of genes associated with amino acid related enzymes, DNA repair and recombination proteins, aminoacyl tRNA biosynthesis, and peptidases after D3. With regard to fungi, we found that there were 51 common genera at day 0 (D0), D3, D14, D28, and D56, including mainly *Cryptococcus*, *Aspergillus*, and *Caecomyces*. *Aspergillus* occupied approximately 47% at day 0, but then it decreased from day 3 to day 14. This study indicates that the core microbes of rumen emerged shortly after birth, but the abundance was very different from the core genus of the adult rumen. In addition, we also report a detailed scheme of the bacterial and fungal colonization process in rumens and propose three distinct stages during the rumen colonization process in pre-weaning goats, which will offer a reference for the development of milk substitutes for small ruminants.

## 1. Introduction

Ruminants have a complex and dynamic ecosystem composed of the rumen microbiota, which interacts with each other and has a symbiotic relationship with the host, contributes to the fermentation of the ingested plant materials, and provides energy for host from the breakdown of plant cell walls [1]. The effects of diet, age, environment, and host on the rumen microbial population have received much attention due to the accessibility offered by high-throughput sequencing [2,3,4]. With increasing age, the rumen microbes gradually play a more important role in the nutrient metabolism of ruminants. The period from birth to weaning is particularly important for an understanding of the rumen microbial maturation process since it is essential for ruminants that the rumen microbes are able to digest the fibers after weaning [5]. Therefore, characterization of the overall rumen bacterial and fungal populations in pre-weaned ruminants is needed.

From birth, microbes quickly colonize the gastrointestinal tract and breast milk can both drive and regulate their community structure and function [2]. The reticulorumen of ruminants cannot digest the milk thoroughly but the majority of it flows to the abomasum via an esophageal groove [1,6]. However, the microbial colonization begins in developing rumen immediately after birth and it undergoes a dramatic change during the first few months of life until weaning [7]. In addition, the microbiota maternal-infantile transmission also plays an important role in the establishment and development of infantile gastrointestinal tract microbiota. For instance, the microbiota of vaginally-delivered infants resembles that of their own mother’s vaginal microbiota [8], which is often dominated by *Lactobacillus*, *Prevotella*, or *Sneathia* [9,10]. The rumen is colonized by microbes originating from sources such as skin, milk, and environment [5]. The process of microbial colonization has been described as a co-evolution process due to the interaction between host and microbes [11,12]. Thus, a clear understanding of the colonization process at each developmental stage is essential to ensure the healthy growth of the pre-weaning goat.

Previous studies have characterized bovine rumen bacterial communities from birth to adulthood through the analysis of 16S rRNA gene sequencing and reported that both diversity and intergroup similarities increased with age [2]. The bacterial changes that occur in the rumen ecosystem after birth are essential for mature rumen function [2]. In an earlier study, we analyzed the goat rumen microbial community from 80 to 110 days old, and reported that the colonization of the microbiota correlated with their function in the rumen [13]. A study also demonstrated the sequential dynamics of ruminal epithelial bacterial colonization in goats at the early stage (day 0, 7, 28, 42, and 70) [14]. However, the primary ruminal bacteria, especially fungal communities which colonized shortly after birth, and the changes within these communities at different growth stages remain largely unknown. Here, we used amplicon sequencing to examine the colonization of rumen microbes in pre-weaning goats at eight time points (day 0 (D0) to D56). The aim of this study was to determine whether the development stage has a significant influence on bacterial and fungal communities. The identified temporal specificity microbes in this study will benefit the optimization of feeding strategies to ensure the rumen health of goat kids.

## 2. Materials and Methods

### 2.1. Animal Handling and Sample Collection

All animal handling protocols were approved by the Northwest A&F University Animal Care and Use Committee (approved ID: 2014ZX08008002). All Shaanbei cashmere goats used in this study have been described previously [15,16]. Briefly, after birth, the male Shaanbei cashmere goat kids were selected and randomly divided into eight age groups (0, 3, 7, 14, 21, 28, 42, and 56 days, *n* = 3 for each group) according to the sampling time point, all Shaanbei cashmere goats were not subjected to any artificial lighting and temperature control, and were kept at natural temperature and light. For the 0-day-age group, newborn kids were separated from the ewes and euthanized immediately before they were suckled. In other groups, kids were housed together with their mothers in the same pen where they were solely fed with colostrum (0 to 3 days) or raw milk until 25 days of age. The kids were allowed access to complete formula granulated feed (Appendix A) from 25 days in addition to breast milk. By the respective deadline, the kids of each age group were sacrificed without prior feeding. Kids were euthanized via injection of thiopental (0.125 mg/kg of body weight; all animals slaughter weight included: D0, 2.78 ± 0.29; D3, 3.92 ± 0.39; D7, 4.63 ± 0.27; D14, 5.45 ± 0.49; D21, 7.07 ± 1.13; D28, 7.87 ± 0.49; D42, 10.32 ± 0.97; and D56, 15.45 ± 1.44 kg) and potassium chloride (5 to 10 mL). The ruminal digesta was collected in 10 mL cryopreservation tubes using sterile spoon, clear and viscous fluid was collected from the rumen of newborn kids, each sample was immediately stored at −80 °C prior to analyses.

### 2.2. DNA Extraction, PCR Amplification, and Illumina MiSeq Sequencing

Total DNA was extracted from rumen fluid samples using a E.Z.N.A.^®^ Stool DNA Kit (Omega Bio-Tek, Norcross, GA, USA). The V3 to V4 hypervariable region of the 16S rRNA gene was amplified by PCR using the following thermocycling protocol: 95 °C for 2 min, followed by 30 cycles at 95 °C for 20 s, 55 °C for 30 s, and 72 °C for 30 s, followed by a final extension at 72 °C for 5 min. The primers used to amplify the sequences were as follows: V338F (5′-ACTCCTACGGGAGGCAGCAG-3′) and V806R (5′-GGACTACHVGGGTWTCTAAT-3′). For the analysis of the fungal microbiota, the samples were PCR amplified using the protocol: 95 °C for 3 min, followed by 35 cycles at 95 °C for 30 s, 55 °C for 30 s, and 72 °C for 45 s, and a final extension at 72 °C for 10 min. The following primers were used for amplification [17]: ITS1F (5′-CTTGGTCATTTAGAGGAAGTAA-3′) and 2043R (5′-GCTGCGTTCTTCATCGATGC-3′). As for the bacterial and fungal populations, an eight-base molecular barcode was added to each sample. PCR reactions were performed in triplicate for 20 mL reactions consisting of 2 mL of 10× FastPfu Buffer, 2 mL of 2.5 mM dNTPs, 0.8 mL of each primer (5 mM), 0.2 mL of FastPfu Polymerase (Transgene, Beijing, China), 200 nM of each primer (Majorbio, Shanghai, China), and 10 ng of template DNA. Gel fragments of correct size was excised and purified using a AxyPrep DNA gel extraction kit (Axygen, Union City, NJ, USA). Purified amplicons were pooled in equimolar and paired-end sequenced (2 × 300 bp) on an Illumina MiSeq platform according to standard protocols. Samples paired-end sequencing by a certified sequencing provider (Majorbio, Shanghai, China).

### 2.3. Data Analysis

Raw fastq files were de-multiplexed, and quality-filtered using QIIME (version 1.9.1) [18] with the following criteria: 300 bp reads were truncated at any site with an average quality score <20 over a 50 bp sliding window; truncated reads shorter than 50 bp were discarded; exact barcode matching; two nucleotide mismatch in primer matching; reads containing ambiguous characters were removed; and sequences with at least a 10 bp overlap were assembled according to their overlap sequence. Reads which could not be assembled were discarded. On the basis of the overlapping sequences, the paired-end reads were merged into a single read. Operational taxonomic units (OTU) were clustered using UPARSE (version 7.1) at least 97% similarity [19,20]. The resulting microbial communities were used for comparison of similarity or dissimilarity between different sample groups, analysis of the relationships between microbial communities and the environment [15]. Principle coordinate analysis (PCoA) graphs were plotted using R Version 2.15.3 [21]. CowPi was used to predict the functional potential of a rumen microbiome using the 16S rDNA data from a previous study [22]. R software was used to analyze the difference between beta diversity index groups. Parametric and non-parametric tests were performed. More than two groups, and the Wilcox test of the Agricolae package was used (* 0.01 < *p* ≤ 0.05, ** 0.001 < *p* < 0.01, *** *p* ≤ 0.001). All data were analyzed using the free online platform of Major I-Sanger Cloud Platform (www.i-sanger.com).

### 2.4. Ethical Statement

The study was approved by the Institutional Animal Care and Use Committee of the Northwest A&F University under permit number 2014ZX08008002.

## 3. Results

### 3.1. Diversity and Commonality Analysis

To characterize the rumen microbiota, twenty-four rumen samples from 24 kids at different ages were collected. The 16S rRNA gene sequencing of the samples yielded 913,234 quality sequences, including an average of 38,301 reads per sample. The total number of OTUs observed was 1592 (Table 1). For fungi, the ITS sequencing of the samples yielded a total of 881,214 quality sequences, including an average of 36,717 reads per sample. The total number of observed OTUs was 1212 (Table 1). The rarefaction curves indicated that the rumen sampling effort provided enough sequencing coverage to accurately describe both the bacterial and fungal composition of each group (Appendix A). Alpha diversity analysis indicated that age affected ACE, Chao1, and Shannon diversity. These indices showed an increased tendency from D0 to D28 (Student’s t-test, *p* > 0.05) and then decreased after D28. The Simpson index decreased with age (Student’s t-test, *p* = 0.01). This result indicates that the diversity of the rumen microbial community at 21 and 28 days were higher than at other time points examined.

For bacteria, the Venn diagram found that 36 common genera coexisted at D0, D3, D14, D28, and D56 (Figure 1A), including mainly *Bacillus*, *Alloprevotella*, *Bacteroides*, *Prevotella_1*, *Lactococcus,* and *Ruminococcaceae_NK4A214*. The number of particular genera presented at 0, 3, 14, 28, and 56 days was 38, 6, 8, 10, and 14, respectively (Figure 1A). The result also showed the proportion of shared genera between D0, D3, D14, D28 and D56 groups, in which all of the genus *Bacillus*, *Alloprevotella*, *Bacteroides*, *Prevotella_1,* and *Lactococcus* were more than 5% (Figure 1B). For fungi, the Venn diagram showed that 51 common genera existed at D0, D3, D14, D28, and D56 (Figure 2A), including mainly *Cryptococcus*, *Aspergillus*, *Caecomyces*, *Lasiosphaeriaceae* family, *Davidiellaceae* family, *Alternaria,* and *Piromyces*, and we found the abundance of these core genera more than 4% (Figure 2B). In addition, the number of fungal genera present only at 0, 3, 14, 28 and 56 days was 19, 25, 17, 22 and 27, respectively (Figure 2A). This indicates that the core microbes of rumen existed shortly after birth and that the abundance is very different from the core genus of the adult rumen.

### 3.2. Temporal Succession of Ruminal Bacterial Community at Pre-Weaning

To investigate the similarity and difference of the rumen bacterial composition of different age groups, unweighted Bray–Curtis distances were calculated using the OTU abundance. The distances between samples at different ages were clustered into the following six stages: D0, D3, D7, D14, D21–D28, and D42–D56 (Figure 3A). To further investigate the temporal colonization progress of the bacterial community at different stages, the bacterial abundance between different age groups was compared at the phylum level. Ten bacterial phyla were identified in rumen fluid samples (Appendix A). The majority of the sequences obtained belonged to Firmicutes (80.49%) and Bacteroidetes at D0 (6.72%) (Appendix A). However, after D0, the population of Bacteroidetes rapidly expanded, accounting for more than 50% of all observed bacteria, indicating it was the dominant phylum (Figure 3C). The relative abundance of Firmicutes remained constant at approximately 20% (Appendix A). Proteobacteria and Fusobacteria gradually decreased with age, particularly after D28, when the feed was switched to roughage. While the relative decrease in the Fusobacteria population was gradual, the decrease of Proteobacteria, was significant from D3 to D7 (Figure 3C). In addition, Fusobacteria and Spirochaete alternated before and after the age of D14 (Figure 3C). To investigate the assignment of dominant bacteria in rumen samples of different ages, three-cluster model finds OTU60 (g_*Bacillus*), OTU946 (*s_Prevotella_heparinolytica*), and OTU286 (*f_Bacteroidales_BS11_gut*) dominated clusters (Figure 3E). These results indicate that the bacterial colonization process with different functions showed three distinct time periods, which were closely related to the differences in diet (i.e., from colostrum to roughage).

At the genus level, a total of 66 genera were identified in the ruminal fluid. At D0, sequencing results indicated the major bacterial genera as *Bacillus*, *Lactococcus*, and *Streptococcus*, the proportion of *Bacillus* was as high as 55.55% (Figure 4A, additional files Appendix A, additional files Appendix A). Through the phylogenetic tree, we found that *Bacillus* and *Lactococcus* have a closer genetic relationship than other bacteria, these exists as a core genus after the animal is born. However, after D0, the relative abundances of these core genera gradually decreased to undetectable levels by D21 (Figure 4B). The proportion of *Bacteroides*, *Porphyromonas*, *unclassified_Ruminococcaceae*, *Bibersteinia*, and *Fusobacterium* at D3, D7, and D14 were higher than at all other ages (Figure 4B). In addition, *Bacteroidales_BS11* and *Ruminococcus_NK4A214* were higher at D14, D21, and D28. *Prevotella_1*, *Treponeme*, *Prevotellaceae* family, and *Porphyromonas* increased from D28, but their relative abundance prior to D21 was negligible (Figure 5). In summary, the results indicate temporal-specificity of the bacteria at the three stages of D0–D14, D14–D28, and D28–D56.

### 3.3. Temporal Succession of Ruminal Fungal Community at Pre-Weaning

Fungi represent an important component of the eukaryotic microbiome in the rumen. To investigate the colonization progress of the fungal community at different stages, the unweighted Bray–Curtis distances were calculated using OTU abundance. The distances between rumen samples of different age groups were divided into two stages, D0 to D7 and D14 to D56 (Figure 3B). Enterotype clustering was performed at the OTU level, two-cluster model finds OTU30 (*s_Aspergillus_flavus*), and OTU743 (*s_Orpinomyces_sp*) dominated clusters (additional files Appendix A). These results indicated that the fungal colonization time is shorter than that of bacteria.

To analyze the process of fungal colonization at different classification levels with age, we compared the fungal abundance between different age groups at the phylum level. Five fungal phyla were identified in digested samples (Figure 3D). The majority of these sequences belong to Ascomycota and Neocallimastigomycota. The relative abundance of Ascomycota accounted for 74.11% at D0 (*F* = 0.240), whereas Basidiomycota accounted for 11.71% (*p* = 0.639), and unclassified Fungi accounted for 13.82% (*p* = 0.047) (additional files Appendix A). After D0, Neocallimastigomycota quickly became the predominant phylum, increasing to more than 60% of all fungi with increasing age of the goats. The relative abundance of Ascomycota remained steady at approximately 30% of the fungal phyla (Figure 3D and additional files Appendix A). At D3, D7, and D14, Basidiomycota was the predominant phylum (Figure 3D).

At the genus level, 41 fungal taxa were identified in the ruminal fluid. At D0, the major components were *Aspergillus*, *Guehomyces*, and *Cladosporium* (Figure 4C and additional files Appendix A). *Aspergillus* was the most abundant genus with levels as high as 36.03% of the total population. However, the proportion of *Aspergillus* decreased in a stepwise manner after D0 (Figure 4C). The proportions of *Caecomyces* and *Lasiosphaeriaceae_unidentified* were higher at D14, D21, and D28 as compared to other age groups (Figure 4C). However, these were observed at only very low proportions after D28. Moreover, the proportions of *Alternaria*, *Davidiella*, *Cladosporium*, *Cryptococcus*, and *Selenomonas_1* remained constant from D0 through to the end of the study period (additional files Appendix A). The proportion of *Neocallimastigaceae_unclassified* gradually increased from D21 to the end of the study period. The relative abundance of *Orpinomyces* peaked at both D28 and D42 (additional files Appendix A).

### 3.4. Predicted Molecular Functions of the Bacterial Microbiota

In this study, CowPi was used to predict the functional gene pathways in the rumen bacteria of goats. The comparisons were made at eight different ages beginning with goat kids through the pre-weaning period of goat rumen samples. The PCoA analysis found that the microbiota function is quite special at D0, the microbiota functions are similar from D14 to D28, and the functions are similar after 28 days of age (Figure 6A). The proportion of amino acid related enzymes; starch and sucrose metabolism; alanine, aspartate and glutamate metabolism; and glycine, serine, and threonine metabolism rapidly increased from D0 to D14, and remained steady after D14 (Figure 6B). The proportion of pyruvate metabolism and glycolysis/gluconeogenesis rapidly decreased after D0 (Figure 6B). On the basis of the excessive metabolic pathway annotation, metabolic pathways were selected for further comparison using the following criteria: the main metabolic pathways included transporters, two-component system, ABC transporters, transcription factors, DNA repair and recombination proteins, purine metabolism, other ion-coupled transporters, ribosome, and peptidases (the relative abundance was more than 2%, Appendix A). Heatmaps showed that the average relative abundance of functional pathways of eight different ages increases in the abundance of genes associated with two-component system, transcription factors, transporters, ABC transporters, and other ion coupled transporters in the rumen at 0-day-old goats (Figure 6C), in addition to increases in the abundance of genes associated with amino acid related enzymes, DNA repair and recombination proteins, aminoacyl tRNA biosynthesis, peptidases after D3 (Figure 6C).

## 4. Discussion

It has been well established that the rumen microbiota aids the host in the extraction of valuable nutrients from plant-based feed, as well as preventing colonization by pathogenic microbes [7]. Examination of the rumen microbiome is often used to assess the health status of the host animal [23,24,25]. At present, reports that examine the rumen fungal community composition of pre-weaning goats are poor, especially exploring the rumen fungal community of newborn goats. To better understand the microbiota of the rumen in pre-weaning goats, both the composition and relative abundance of rumen bacteria and fungi at D0 to D56 were studied using high-throughput sequencing. The effects of microbial community colonization were further examined by analyzing the predominant metabolic pathways detected in relation to goat age using gene function prediction analysis.

In this study, the predominant rumen bacterial communities consisted of Bacteroides, Firmicutes, and Proteobacteria. These findings are consistent with previous reports [26,27,28]. The proportion of Firmicutes was observed to be as high as 80% on D0, while the relative abundance of Bacteroides comprised approximately 8%. After D0, Bacteroides were quickly established as the predominant bacterial community, accounting for approximately 50% of all bacteria, and remained largely steady throughout the entire study period. Interestingly, the rumen microbiota was dominated by Firmicutes, while *Bacillus* and *Lactococcus* were predominant at the genus level at D0, the observed proportion of *Bacillus* reached a maximum of 55.55%. After nursing, the predominant bacterial phylum observed was Bacteroidetes, while the proportions of *Bacillus* and *Lactobacillus* were very low. A previous study found that *Lactobacillus* can degrade glycans and carbohydrates, and make a major metabolic contribution to their host [29], and *Bacillus* and *Lactobacillus* can enhanced epithelial barrier function, enhanced mucosal IgA response, direct antagonism of pathogens, competitive rejection of pathogens, prevention of apoptosis, production of anti-inflammatory cytokines, and downregulation of pro-inflammatory pathways [30,31]. However, breastfeeding provides a mixture of nutrients, probiotics, and antibacterial agents that cause major changes in the rumen microbiota and selectively shape the growth and colonization of other dominant microbes [32]. This is an important reason for the lower proportion of *Bacillus* and *Lactobacillus* after nursing.

Several investigators have reported the presence of microbes in the meconium [33,34], although their origin could not be clearly pinpointed. It has been suggested that the meconium microbiota may be derived from the amniotic fluid ingested in utero [35]. We also suggested that the D0 rumen microbiota of goat kids may be derived from the ingestion of amniotic in utero. It has been previously reported that the percentage of *Bacillus* in amniotic fluid was high [35,36,37], which was consistent with the results of this study. In addition, several *Bacilli* in the gut of the giant panda (*Ailuropoda melanoleuca*) have been reported to play an important role in the decomposition of fiber and exhibited growth inhibitory effects on several common intestinal pathogens [38]. The transcription of *Bacillus subtilis* HH2 cultured in the presence of various carbon sources was studied using transcriptome sequencing techniques. The different carbon sources were observed to activate specific transcriptional controls (such as elevated expression of cellulose and the reduction of non-essential protein synthesis to save energy) in the giant panda [39]. These studies reported *Bacillus* as a suitable candidate for micro-ecological preparations. For example, *Bacillus* plays a role in the secretion of cellulose, which aids in the digestion of cellulose by animals and in the production of a variety of antimicrobial peptides that help animals to maintain the normal gut microbiota.

At the genus level, a very interesting phenomenon was observed with respect to the rumen bacteria. The proportions of *Prevotella_1*, *Treponema_2*, and *Rikenellaceae_RC9* were undetectable prior to D14, but these bacterial genera were maintained at specific relative abundance for each genus, starting at D14 through to the end of the study period. These results suggest that the colonization of these genera may be related to their abilities to degrade cellulose, as well as in the metabolism of the resulting carbohydrates. It has been suggested that an important function of *Prevotella_1* was to aid in the digestion of feed proteins [40]. Similarly, Jami et al. reported that *Prevotella* colonization of the rumen occurred gradually until it became the predominant bacterium in cattle on high fiber diets [2]. Additionally, at the genus level, the proportion of *Bacteroides*, *Porphyromonas*, *Fusobacterium*, and *Neisseriaceae* family was also undetectable at D0 but increased from D3 to D14. Several investigators have reported [41,42] the presence of microbes in the colostrum and indicated that colostrum (vaginally delivered vs. cesarean delivered) had significantly lower abundance of *Pseudomonas sp*, *Staphylococcus sp*, and *Prevotella sp*. In addition, several studies have reported [43,44,45] that bacteria commonly found in breast milk include *Streptococcus*, *Veillonella*, *Gemella*, *Enterococcus*, *Clostridia*, *Bifidobacterium*, *Lactobacillus*, *Sphingomonas*, *Serratia*, *Escherichia*, *Enterobacter*, *Ralstonia*, *Bradyrhizobium*, *Propionibacterium*, *Actinomyces,* and *Corynebacterium*. Therefore, the intake of colostrum greatly shapes the rumen microbial community. The colostrum microbiota can enhance immune development through microbial ligands, both nutrient metabolism and absorption, an improve intestinal barrier function [45]. Previous studies have reported that it can promote different rumen microbial colonization of natural and artificial feeding management before weaning but not differences in gene expression levels at the rumen epithelium of newborn goats, and *Coriobacteriaceae* and *S24–7* families changed between feeding system and age [46]. Ruminal diet intervention of sheep and goats in pre-weaning is important for rumen development, especially after temporal microbial colonization. De Barbieri et al. study found that an intervened diet of maternal ewes and lambs in the neonatal period can alter the ruminal microbes’ communities, especially inoculating the rumen microbes during the first weeks of a lamb’s life [47]. Previous studies have focused on the relationships among the microbiota of the dam’s vagina, skin, and colostrum to the early successional development of the calf microbiome [5]. A previous study has provided the first description of the maternal influence on the early ecological dynamic of microbiota throughout the calf GIT, and reflected the important influences of microbiota from the skin of the dam’s udder [5]. However, after D14, these genera gradually decreased in abundance within the rumen. According to the results of the functional prediction analysis, these bacterial genera may assist the host with the digestion of maternal milk during the time of exclusive nursing. Over time, these bacterial genera were gradually replaced by other bacteria, which are capable of digesting structural carbohydrates (i.e., cellulose, hemicellulose, and xylogen) into soluble saccharides. In line with our recent study, in which we investigated the colonization of reticulum, omasum, and abomasum microbes in pre-weaned goats [15], we revealed that developmental stages have significant effects on microbiota communities in different stomach compartments of goats.

In ruminants, the rumen performs many physiological functions. CowPi was used to predict bacterial gene functions and the results indicated that the most abundant functional categories corresponded to the functions of transporters, two-component system, ABC transporters, transcription factors, DNA repair and recombination proteins, purine metabolism, and other ion-coupled transporters, ribosome and peptidases. These findings are consistent with general metabolic functions that are essential for microbial survival [48]. Previous studies focused heavily on rumen bacterial diversity and function [49,50]. However, the role of rumen fungi remains largely unexplored. Previous studies have reported that the rumen contained higher proportions of Ascomycota, Basidiomycota, and Neocallimastigomycota [51,52]. In this study, *Aspergillus* accounted for approximately 47% of the fungal population at D0 but declined to undetectable levels from D3 to D14, however, the amniotic fluid culture tested positive for *Aspergillus flavus*. Therefore, we suggest that rumen fungi may be derived from swallowed amniotic fluid of goat kids at D0 [53,54].

This is supported by the apparent lack of *Neocallimastix_sp*, *Orpinomyces_sp*, *Caecomyces*, and *unclassified_f_Neocallimastigaceaae* in the rumen prior to D14. The proportion of these genera gradually became the predominant fungal genera after D14. *Neocallimastix_sp* and *Orpinomyces_sp* are the most important fiber-degrading fungi in the rumen of ruminants. A previous study reported an abundance of Orpinomyces in the rumen of bovines that received a high-protein and hay-based diet [55]. In addition, a previous study has shown that *Orpinomyces* represented the highest lignocellulose-degrading enzymatic activity [56]. As a result of their filamentous growth and cutinize activity, *Neocallimastix* showed the highest enzymatic activity [55]. In addition, it has been suggested that the fungal community may be determined by diet and other host factors [57,58,59]. Therefore, with the change of diet, the fungal community of the rumen likely undergoes dramatic changes. At the species level, we observed that the fungi of *Orpinomyces_sp, Caecomyces communis*, *Neocallimastix_sp*, *Piromyces_sp*, and *unclassified_g_Neocallimastix* became the predominant microbes after D14, while these species were undetectable prior to this time point. Therefore, we conclude that these fungi may be involved in the digestion of feed fiber in rumens.

## 5. Conclusions

In summary, we present, here, that the rumen bacterial and fungal community profiles varied considerably throughout the pre-weaning stage of life in goats. Namely, D0–D14 is a time for significant microbe development, D14–D28 underwent a substantial transition of microbiota colonization, whereas the rumen bacterial colonization is stabilized from D28–D56. We further show that the bacterial community is associated with an enhanced immunity after birth, and then transited to digestion and metabolism. Together with our recent results in reticulum, omasum, and abomasum in pre-weaned goats [15], we provided a comprehensive landscape of bacterial and fungal community in small ruminants at early stages. Our results will provide new insights into the colonization of bacterial and fungal communities in goats that may be useful in the development of starters for small ruminants.

## Figures and Tables

**Figure 1 animals-09-01028-f001:**
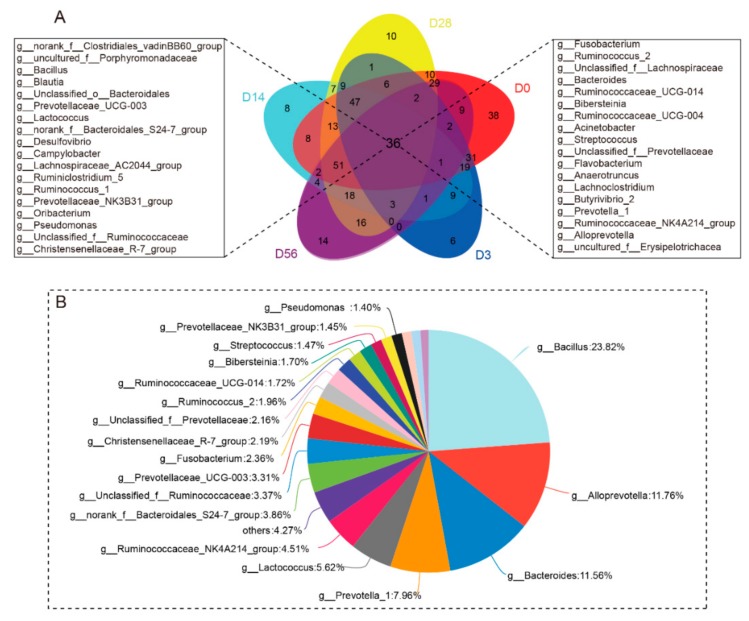
Bacterial composition similarity and overlap in genera level between eight different age samples. (**A**) Venn graph analysis. Different colors represent different groups, overlapping parts represent genus that are common among multiple groups, parts of not overlap represent genus that are specific to the group, and numbers indicate the number of genus corresponding. (**B**) Proportional distribution belongs to the common genus throughout the age. Different colors indicate different genus, and the pie chart area indicates the number of genus in the total number of genus.

**Figure 2 animals-09-01028-f002:**
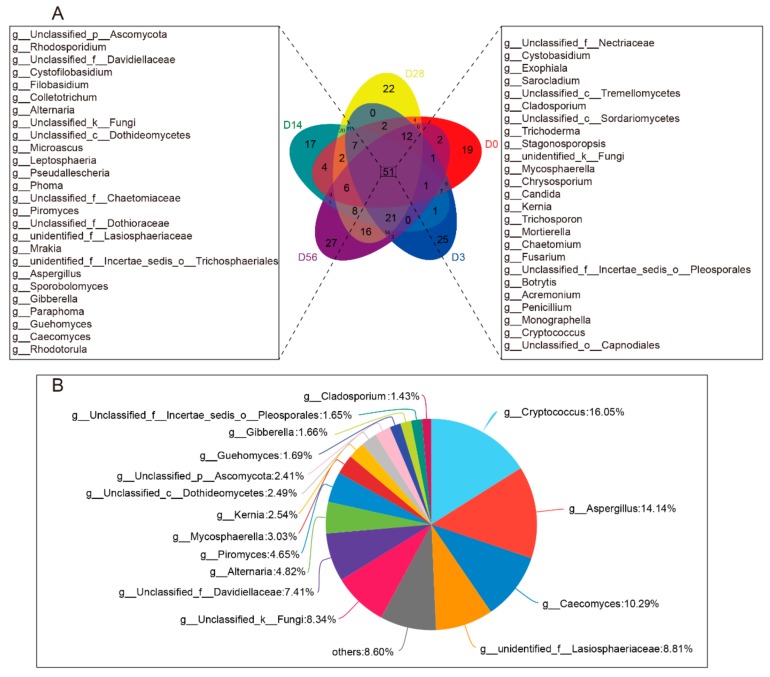
Fungal composition similarity and overlap in genera level between eight different age samples. (**A**) Venn graph analysis. Different colors represent different groups, overlapping parts represent genus that are common among multiple groups, parts of not overlap represent genus that are specific to the group, and numbers indicate the number of genus corresponding. (**B**) Proportional distribution belongs to the common genus throughout the age. Different colors indicate different genus, and the pie chart area indicates the number of genus in the total number of genus.

**Figure 3 animals-09-01028-f003:**
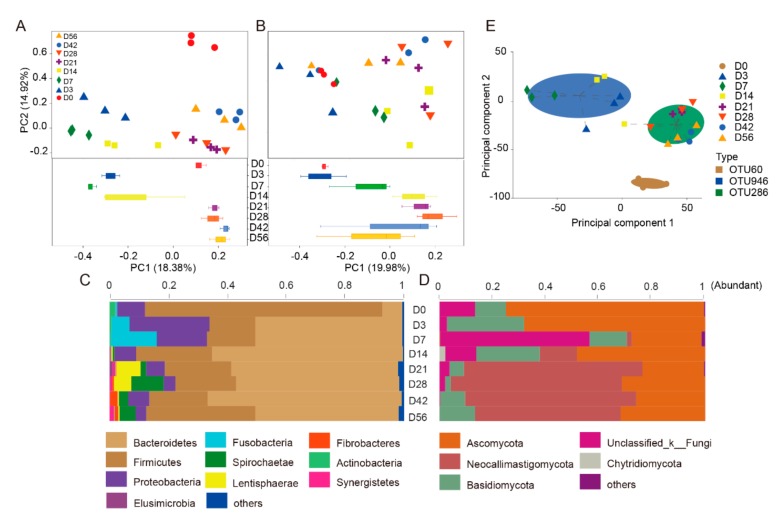
Bacterial and fungal community dissimilarities between eight different age samples. (**A**) Principal coordinate analysis at the OTU level of the community structure in bacteria of the goat rumen. (**B**) Principal coordinate analysis at the OTU level of the fungal community structure of the goat rumen. Different colors represent different age groups. The compositions of all bacteria (**C**) and fungi (**D**) at the phylum level of goat rumen of different age groups. (**E**) The classification of dominant bacterial populations in different samples was studied mainly via statistical clustering.

**Figure 4 animals-09-01028-f004:**
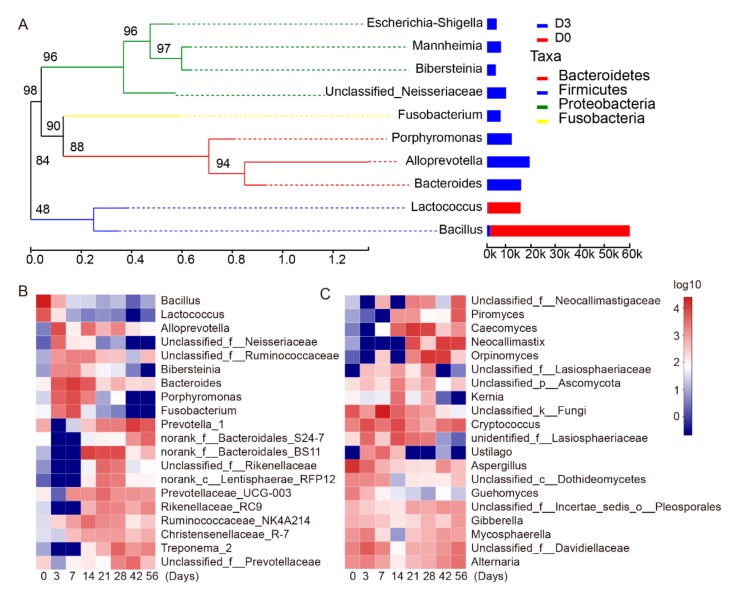
Relative abundance of bacterial and fungal genera for eight different age samples. (**A**) Phylogenetic tree on genus level. Each branch in the phylogenetic tree represents genu, and the length of the branch is the evolutionary distance between the two genera; the right column shows the relative proportion of reads that belong to different species in each group. (**B**) Compositions of all bacteria based on genus sequences in goat rumens in different age groups. (**C**) Compositions of all fungi based on genus sequences in the goat rumens in different age groups.

**Figure 5 animals-09-01028-f005:**
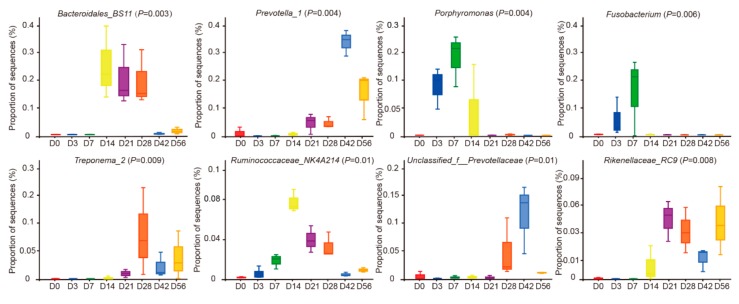
Colonization of specific bacteria in the rumen. Test of significance difference of goat ruminal bacteria of different age groups.

**Figure 6 animals-09-01028-f006:**
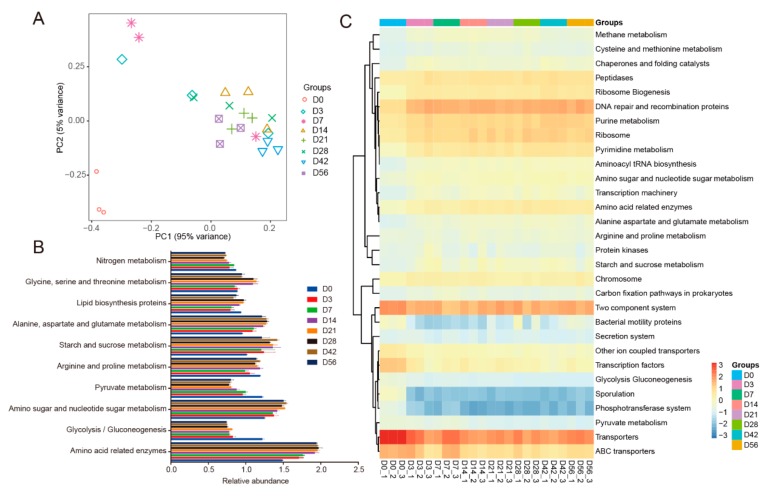
Metagenomic functional predictions using CowPi. (**A**) The PCoA analysis of different age groups in KEGG metabolic pathways. (**B**) Variations in KEGG metabolic pathways in functional bacterial communities throughout the rumen of goat kids. (**C**) Heatmaps show the average relative abundance of functional pathways of eight different ages.

**Table 1 animals-09-01028-t001:** Valid sequences and alpha diversity.

Days	Bacterial						Fungi					
	Reads	OTU	Ace	Chao	Shannon	Simpson	Reads	OTU	Ace	Chao	Shannon	Simpson
D0	23,242	364.33 ^abc^	493.67 ^abc^	498.33 ^ab^	2.40 ^d^	0.33 ^a^	28,995	123.67 ^b^	129.00 ^c^	132.33 ^c^	2.8	0.18
D3	23,242	151.00 ^d^	207.00 ^d^	196.33 ^c^	3.05 ^cd^	0.13 ^b^	28,995	176.00 ^ab^	191.00 ^c^	190.00 ^c^	2.79	0.15
D7	23,242	183.00 ^d^	226.00 ^d^	226.67 ^c^	2.87 ^cd^	0.12 ^b^	28,995	227.00 ^ab^	260.33 ^bc^	258.67 ^abc^	2.46	0.25
D14	23,242	269.67 ^cd^	368.67 ^cd^	357.00 ^bc^	3.13 ^bcd^	0.10 ^b^	28,995	206.33 ^ab^	245.33 ^bc^	253.67 ^abc^	2.44	0.25
D21	23,242	519.33 ^a^	626.67 ^ab^	637.67 ^a^	4.06 ^ab^	0.05 ^b^	28,995	285.00 ^a^	360.67 ^ab^	359.67 ^ab^	3.11	0.11
D28	23,242	503.33 ^ab^	656.00 ^a^	641.67 ^a^	4.07 ^ab^	0.05 ^b^	28,995	314.33 ^a^	431.33 ^a^	403.67 ^a^	2.9	0.13
D42	23,242	348.67 ^bc^	447.67 ^bc^	457.33 ^ab^	3.64 ^abc^	0.06 ^b^	28,995	179.00 ^ab^	234.00 ^bc^	217.00 ^bc^	2.38	0.22
D56	23,242	512.00 ^a^	590.67 ^ab^	595.67 ^a^	4.42 ^a^	0.04 ^b^	28,995	202.33 ^ab^	260.00 ^bc^	251.67 ^abc^	2.37	0.22
SEM		32.28	38.26	38.95	0.16	0.02		18.31	22.11	22.577	0.145	0.025
*p*-value		0.24	0.24	0.21	0.04	0.01		0.02	0.09	0.1	0.16	0.15

Note: ^a,b,c,d^ Values in the same column with different superscripts differ significantly (*p* < 0.05); SEM represent standard error of mean. OTU, operational taxonomic units

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
