# Peer review of "Maturation of the Goat Rumen Microbiota Involves Three Stages of Microbial Colonization"

_animals, 2019, doi:10.3390/ani9121028_

Round 1

Reviewer 1 Report

Main concerns:

Predict molecular function of the bacterial microbiota in rumen should be done with CowPI and not with PICRUSt. Please rewrite results and conclusions accordingly.

References are mainly from humans and mice studies. However, key references in ruminants are missing. I recommend using references from CJ Newbold and DR Yañez-Ruiz´s groups for discussion. They published a similar work comparing two different feeding management before weaning in goats that was in agreement with this study.

Minor points:

Information about animals and farm conditions are missing: breed, sex, temperature, hours of light,..

Al least provide the platform used for sequencing

Flora should be removed from the text. Microbiota should be used instead.

Revise sentence in line 339, because "this study" is not clear. Do you refer to goats or calves?

Author Response

MS title: Maturation of the Goat Rumen Microbiota Involves Three Stages of Microbial Colonization

Author: Zhang K, et al.

MS number: animals-630355

Responses to editors and reviewers’ comments

Dear Editors and Reviewers:

Thank you for your letter and for the reviewers’ comments concerning our manuscript. Those comments are all valuable and very helpful for revising and improving our paper, as well as the important guiding significance to our researches. We have studied comments carefully and have made correction which we hope meet with approval. Revised portion are marked in red in the paper. The main corrections in the paper and the responds to the reviewer’s comments are as flowing:

Reviewer #1 (Remarks to the Author):

Q1: Predict molecular function of the bacterial microbiota in rumen should be done with CowPI and not with PICRUSt. Please rewrite results and conclusions accordingly.

RE: Good suggestion! I have revised the predict molecular function used CowPi, and rewrite results and conclusion. See line 263-289.

Q2: References are mainly from humans and mice studies. However, key references in ruminants are missing. I recommend using references from CJ Newbold and DR Yañez-Ruiz´s groups for discussion. They published a similar work comparing two different feeding management before weaning in goats that was in agreement with this study.

RE: Thanks a lot. We have added the discussion according to you recommend references. See line 355-362.

Minor points:

Q3: Information about animals and farm conditions are missing: breed, sex, temperature, hours of light

RE: Good suggestion! We have revised the parts of farm condition, all Shaanbei cashmere goats were not subjected to any artificial lighting and temperature control, and were kept at natural temperature and light. please see line 89-105.

Q4: Al least provide the platform used for sequencing.

RE: We have provided the platform used for sequencing. Purified amplicons were pooled in equimolar and paired-end sequenced (2×300 bp) on an Illumina MiSeq platform according to standard protocols. Please see line 121-123.

Q5: Flora should be removed from the text. Microbiota should be used instead.

RE: We have used “microbiota” instead “flora”.

Q6: Revise sentence in line 339, because "this study" is not clear. Do you refer to goats or calves?

RE: Thanks. We have revised this sentence.

Reviewer 2 Report

The authors provide a good introduction explaining the background and rationale for their work and they have clearly described their experimental objectives.  The materials and methods are generally written clearly and for the most part is appropriate for stated objectives although I list below some minor correction needed. The results are presented objectively. Conceptually and scientifically, this experiment is worthwhile and timely. The manuscript is well written from a grammatical point of view although I list below a few items for the authors’ consideration. 

Comments for authors’ consideration:

L88: Change to “handling”

L88-90: add type of breed. E.g., Dorper sheep or merino sheep etc.

L96: how did you slaughtered animals and add brand name and amounts of drugs used for euthanized. Add animal body weight before slaughter.

L98: add more detail information for rumen fluid collection and procedures from the experimental animals. Did you filtered rumen fluid through cheese clothes or used as-it is.

L148: no italic for number after bacterial species “NK4A214” etc.

L162/L169: no comma after “(B)” in Figs.1 and 2

L264: need a line adjustment after heatmaps….

L373-374: You didn’t measured correlation between microbiome and milk constitutes. So, you

                 may need to change this text.

L546 (reference). Add detail information after Oklahoma State University.

Author Response

MS title: Maturation of the Goat Rumen Microbiota Involves Three Stages of Microbial Colonization

Author: Zhang K, et al.

MS number: animals-630355

Responses to editors and reviewers’ comments

Dear Editors and Reviewers:

Thank you for your letter and for the reviewers’ comments concerning our manuscript. Those comments are all valuable and very helpful for revising and improving our paper, as well as the important guiding significance to our researches. We have studied comments carefully and have made correction which we hope meet with approval. Revised portion are marked in red in the paper. The main corrections in the paper and the responds to the reviewer’s comments are as flowing:

Reviewer #2 (Remarks to the Author):

The authors provide a good introduction explaining the background and rationale for their work and they have clearly described their experimental objectives. The materials and methods are generally written clearly and for the most part is appropriate for stated objectives although I list below some minor correction needed. The results are presented objectively. Conceptually and scientifically, this experiment is worthwhile and timely. The manuscript is well written from a grammatical point of view although I list below a few items for the authors’ consideration.

Comments for authors’ consideration:

Q1: L88: Change to “handling”

RE: We have modified it.

L88-90: add type of breed. E.g., Dorper sheep or merino sheep etc.

RE: Good suggestion! All Shaanbei cashmere goats were used in this study. please see line 89-105.

L96: how did you slaughtered animals and add brand name and amounts of drugs used for euthanized. Add animal body weight before slaughter.

RE: Good suggestion! By the respective deadline, the kids of each age group were sacrificed without prior feeding. Kids were euthanized via injection of thiopental (0.125 mg/kg of body weight; all animals slaughter weight including that, D0: 2.78±0.29, D3: 3.92±0.39, D7: 4.63±0.27, D14: 5.45±0.49, D21: 7.07±1.13, D28: 7.87±0.49, D42: 10.32±0.97, D56: 15.45±1.44 kg) and potassium chloride (5–10mL). please see line 89-105.

L98: add more detail information for rumen fluid collection and procedures from the experimental animals. Did you filtered rumen fluid through cheese clothes or used as-it is.

RE: Thanks. The ruminal fluid digesta was collected into 10 mL cryopreservation tubes using sterile spoon. See line 100-105.

L148: no italic for number after bacterial species “NK4A214” etc.

RE: We have revised it.

L162/L169: no comma after “(B)” in Figs.1 and 2

RE: We have revised it.

L264: need a line adjustment after heatmaps….

RE: We have revised it.

L373-374: You didn’t measured correlation between microbiome and milk constitutes. So, you

may need to change this text.

RE: Good suggestion! We have revised it.

L546 (reference). Add detail information after Oklahoma State University.

RE: Thanks, we have revised it. Fernando, B.R. Metagenomic analysis of microbial communities in the bovine rumen. Ph.D. thesis, Oklahoma State University, Stillwater, 2012.

We tried our best to improve the manuscript and made some changes in the manuscript. These changes will not influence the content and framework of the paper. And here we did not list the changes but marked in red in revised paper.

We appreciate for Editors/Reviewers’ warm work earnestly, and hope that the correction will meet with approval.

Once again, thank you very much for your comments and suggestions.